# Benchmark Dataset for Training Machine Learning Models to Predict the Pathway Involvement of Metabolites

**DOI:** 10.3390/metabo13111120

**Published:** 2023-11-01

**Authors:** Erik D. Huckvale, Christian D. Powell, Huan Jin, Hunter N. B. Moseley

**Affiliations:** 1Markey Cancer Center, University of Kentucky, Lexington, KY 40506, USA; 2Superfund Research Center, University of Kentucky, Lexington, KY 40506, USA; 3Department of Computer Science (Data Science Program), University of Kentucky, Lexington, KY 40506, USA; 4Department of Toxicology and Cancer Biology, University of Kentucky, Lexington, KY 40536, USA; 5Department of Molecular and Cellular Biochemistry, University of Kentucky, Lexington, KY 40506, USA; 6Institute for Biomedical Informatics, University of Kentucky, Lexington, KY 40506, USA

**Keywords:** metabolite, pathway, machine learning, KEGG, kegg_pull, md_harmonize, atom color

## Abstract

Metabolic pathways are a human-defined grouping of life sustaining biochemical reactions, metabolites being both the reactants and products of these reactions. But many public datasets include identified metabolites whose pathway involvement is unknown, hindering metabolic interpretation. To address these shortcomings, various machine learning models, including those trained on data from the Kyoto Encyclopedia of Genes and Genomes (KEGG), have been developed to predict the pathway involvement of metabolites based on their chemical descriptions; however, these prior models are based on old metabolite KEGG-based datasets, including one benchmark dataset that is invalid due to the presence of over 1500 duplicate entries. Therefore, we have developed a new benchmark dataset derived from the KEGG following optimal standards of scientific computational reproducibility and including all source code needed to update the benchmark dataset as KEGG changes. We have used this new benchmark dataset with our atom coloring methodology to develop and compare the performance of Random Forest, XGBoost, and multilayer perceptron with autoencoder models generated from our new benchmark dataset. Best overall weighted average performance across 1000 unique folds was an F1 score of 0.8180 and a Matthews correlation coefficient of 0.7933, which was provided by XGBoost binary classification models for 11 KEGG-defined pathway categories.

## 1. Introduction

Metabolomics is the systematic study of the biomolecules present in a given living system that can be composed of one or more organisms. Metabolomics is often used to study metabolism, the set of life sustaining biochemical reactions occurring in these living systems. These reactions occur in coordinated and regulated pathways, where the product of the previous reaction is the substrate for the next reaction. One or more groups of interconnected or related metabolic pathways form a metabolic pathway category. These pathway categories can be organized in a hierarchy with broader categories being at the top of the hierarchy and more specific categories being lower in the hierarchy [1]. The biochemical reactions and metabolites associated with these pathway categories can be linked in data sources such as the Kyoto Encyclopedia of Genes and Genomes (KEGG) [2,3,4] and MetaCyc [5,6], and PubChem [7] with cross-references to each other or additional resources such as Reactome [8,9].

While such data sources link some metabolites to their pathway involvement, many of these links are missing. The advances in Mass Spectroscopy (MS) and Nuclear Magnetic Resonance (NMR) technologies over the past few decades, especially in terms of sensitivity, have led to a dramatic increase in the amount of metabolomics data being collected and uploaded to databases such as Metabolomics Workbench [10] and MetaboLights [11]. However, at best, roughly 50% of detected compounds can be assigned to a pathway category. Often it is the case that the metabolic roles of these experimentally identified compounds are unknown, since the metabolic network databases, e.g., KEGG and MetaCyc, are grossly incomplete in terms of the known biochemical reactions [1]. This has led to the need to accurately predict the metabolic pathway involvement of unassigned compounds [12].

Several machine learning methods have been developed to predict the pathway involvement of metabolites in the form of mapping metabolites to broad hierarchical pathway categories given the molecular structure of said metabolite. Most notably were the machine learning models trained on the dataset created by Baranwal et al. [13] composed of SMILES representations of metabolic compounds with associated pathway labels. They claim they obtained the SMILES data from KEGG, so we will refer to it as the KEGG-SMILES dataset; however, KEGG does not provide a SMILES representation of their KEGG COMPOUND entries. One of the most recent models trained on the KEGG-SMILES dataset was developed by Du et al. and reports the highest performance compared to past models trained on this dataset [14]. However, Huckvale et al. discovered thousands of duplicate entries in the KEGG-SMILES dataset, rendering the dataset and the results in the publications utilizing this dataset as invalid [15]. Huckvale et al. also pointed out that the description of its creation lacks sufficient detail to recreate it, let alone code that one could simply re-run to re-generate it. Therefore, there is a need for a new benchmark dataset that meets the requirements outlined by Huckvale et al., including transparent details of its creation, data validation including ensuring that all entries represent unique metabolites, code and raw data to reproduce the dataset, and finally scripts and instructions for building off of it as KEGG updates their data. In this work, we present a benchmark dataset, generated from KEGG compound and pathway data, that fulfills all these requirements. We additionally include an analysis correlating the amount of chemical information within metabolites to the reliability in predicting their pathway categories, followed by filtering metabolites with insufficient chemical information for reliable pathway classification.

Beyond the ability to predict the pathway involvement of metabolites, there is also the need to determine which molecular substructures within the metabolites are most important for this prediction. Rather than training black box models, Jin et al. [16] presents an atom coloring method, which generates molecular structure representations of compounds, which, when used as features for a machine learning tabular dataset, we can determine which molecular substructures are most associated with pathway involvement by measuring feature importance. In this work, we use the atom coloring method to generate the benchmark dataset and train three different types of machine learning models i.e., Random Forest (RF), Multi-layer Perceptron (MLP), and eXtreme Gradient Boosting (XGBoost) with feature importance measured for the XGBoost. Thus, we present a benchmark dataset as well as benchmark model performance results from which future publications can build upon, including the capacity to investigate molecular substructure importance in metabolites.

## 2. Materials and Methods

### 2.1. KEGG Data Pull

We used the kegg_pull [17] Python package to download all available entries and associated molfiles from the KEGG COMPOUND database [2,3,4]. As of 3 July 2023, 19,119 compound entries were retrieved from the KEGG database (Figure 1). However, not all of the available compound entries are metabolites and some entries have no molfile associated with them. We used the ‘Metabolism’ section of the KEGG BRITE br08901 hierarchy (https://www.genome.jp/brite/br08901, accessed on 3 July 2023) to determine which of the KEGG compounds were associated with broad metabolic pathways as defined by KEGG. The ‘Metabolism’ section of the br08901 hierarchy contains 12 distinct metabolic pathway category branches, excluding ‘Global and overview maps’ since this is a catchall category. Each of these 12 metabolic pathways then branch out into leaf nodes representing more specific pathway categories.

### 2.2. Dataset Creation

#### 2.2.1. Initial Data Cleaning and Filtering

For the first filtering step in Figure 1, we used functionality developed in kegg_pull [17] to link 6736 compound entries to the pathways they are associated with. However, not all of them are linked to specific metabolic pathways, so we filtered further by taking only those compound entries linking to these specific metabolic pathway leaf nodes. Using this method (second filtering step in Figure 1), we identified 6234 compounds associated with these specific metabolic pathways. We consider these metabolite entries and linked each entry to the pathway categories in the hierarchy level above the leaf nodes, i.e., 1 or more of the 12 broad metabolic pathway categories. We call these links metabolic ‘pathway labels’ of the entry. Next, we downloaded the molfiles corresponding to each metabolite entry from the KEGG compound database, if available. This represented a third filtering step illustrated in Figure 1, since only 6144 metabolite entries had a molfile available.

Next, there were a few pairs of metabolite entries with identical or equivalent molfiles. We handled these duplicate molfiles by merging the pair of compound entries into a single representative entry, keeping only one molfile and creating the union set of their metabolite pathway labels. After de-duplicating the duplicate molfiles (fourth filtering step in Figure 1), 6142 compound entries remained, representing the initial KEGG metabolite dataset.

#### 2.2.2. Filtering by Information Content

However, some compound entries, like KEGG C00001 (i.e., water or H_2_O), contained very few non-hydrogen atoms and thus very little chemical information. With a histogram, Figure 2a illustrates the non-hydrogen atom count across the initial KEGG metabolite dataset. Therefore, we hypothesized that metabolites containing too few non-hydrogen atoms (less chemical information) could not be reliably classified. If this hypothesis were true, we would want to determine an optimal minimum number of non-hydrogen atoms in compounds to use for pathway classification. To both test this hypothesis and determine the optimal minimum non-hydrogen atom count, we investigated how many non-hydrogen atoms result in a Random Forest (RF) model consistently classifying the compound correctly (i.e., determine whether it belongs to one of the 12 pathway labels or not). The RF training algorithm is stochastic, producing a slightly different model over repeat trainings even when trained on the same data. This enabled us to measure the percentage of misclassifications for each compound in the unfiltered (initial) dataset (size of 6142) across 1000 model training/evaluation iterations. For each of the 12 pathway categories, we trained a binary classifier RF model on the unfiltered dataset, the features of which were constructed from the molfiles using the method described below. We determined the misclassification rate of a given compound for each of the 12 pathway categories, averaging the 12 misclassification rates to obtain the overall misclassification rate per compound. We additionally measured the number of non-hydrogen atoms in the corresponding molfile.

Comparing non-hydrogen atom count to the misclassification rate as illustrated in Figure 2b, we see a trend of a higher number of non-hydrogen atoms resulting in less misclassification, with misclassification consistently being 0 towards a non-hydrogen count of 100. However, if we were to filter entries from our dataset with non-hydrogen atom counts near 100, we would filter out the majority of our data (see histogram in Figure 2a).

To balance training a model that is maximally reliable at classifying compounds with being permitted to classify as many compounds as possible, we performed a sliding window analysis with each window being a range of non-hydrogen atom counts (beginning at a count of 0) and the dependent variable being the average misclassification rate of the compounds within the window (compounds with a non-hydrogen atom count within the range). Using a window size of 5, as illustrated in Figure 3a, we see that the average misclassification rate of compounds with non-hydrogen atom counts within a range of 0 and 4 (inclusive) is above 0.04. The next window (between 1 and 5) has an average of almost 0.03 and so on.

Zooming into the first 20 points (Figure 3b), we see more clearly a major drop in the first several windows, reaching a local minimum at the window beginning with a non-hydrogen atom count of 7. This result justified using a non-hydrogen atom count threshold of 7, i.e., our model will neither train nor predict on compounds with a number of non-hydrogen atoms less than a minimum of 7. While there are later windows with rates that go up, selecting the threshold at the first local minimum balances between training a reliable classifier and being permitted to predict on a wider range of compounds (a wider range of non-hydrogen atom count). After filtering compounds with less than 7 non-hydrogen atoms from the unfiltered initial dataset (fifth filtering step in Figure 1), there were 5884 compounds remaining.

#### 2.2.3. Atom Color Feature Generation

The molfiles (in both the filtered and unfiltered dataset) provided the raw data for constructing the chemical features, with each molfile being transformed to a chemical feature vector with associated metabolic pathway labels. To construct chemically-informative features from the molfiles, we used the atom coloring technique introduced by Jin et al. [16,18,19], specifically with the md_harmonize Python package [20] that implements this method. The atom colors corresponding to a particular compound have greater detail with increasing bond inclusivity. With a bond inclusivity of 0 (0-bond-inclusive), atom colors are just the elemental identities of the individual atoms in the compound. The resulting count of these 0-bond-inclusive atom colors is equivalent to their non-hydrogen chemical formula. For example, ethanol has 2 C (carbon) and 1 O (oxygen) atom colors. When increasing the bond inclusivity to 1 (1-bond-inclusive), we obtain atom colors representing all atoms within one bond, e.g., ethanol has C-C, C-[C,O], and O-C atom colors. A 2-bond-inclusive produces atom colors that have an additional bond in a chain, e.g., a C-C-O, C-[C,O], O-C-C for ethanol. More complex compounds can have atom colors with 3 or more bond inclusivity. We generated all possible atom colors up to bond inclusivity of 3 for each molfile, where bond inclusivity beyond 0 excluded hydrogen atoms.

Across all molfiles, there were 23 0-bond-inclusive features with an increasing number of features as feature sets that were concatenated to those with higher bond inclusivity (Table 1). In the resulting dataset, the columns represented a given atom color feature and the rows represented each metabolite, being derived from the corresponding molfile. These atom-coloring features of the dataset were the number of times each atom color appeared in a given molfile. Most features appeared 0 times in a given molfile, but every feature appeared in at least one molfile. In this dataset, we observed duplicate feature vectors, even though the molfiles were not equivalent. This is not surprising for 0-bond-inclusive features, since various compounds share the same molecular formula. Also unsurprisingly, there were duplicate feature vectors with a bond inclusivity of 1 and 2 as well, though the number of duplicates decreased as the bond inclusivity increased. We stopped at a bond inclusivity of 3 since the number of duplicate feature vectors stopped decreasing for a bond inclusivity of 4 and the number of features was becoming unmanageably large (Table 1), meaning it required far too many computational resources to create and process downstream. The remaining duplicates for the 3-bond-inclusive features resulted from factors other than bond inclusivity and needed to be handled manually as detailed below.

The md_harmonize package [20] allows for incorporating information about a compound into its atom colorings in addition to the elemental identity of its atoms. The atom coloring method can be configured to include specific chemical details from the molfile (Appendix A). This enabled us to add additional details such as stereochemistry, bond order, and R groups. While we included R groups (Appendix A) in the atom colors, we replaced the ‘R’ symbol in each with ‘C’ in order to emphasize the known chemical information when training the model rather than exposing the model to unknown R groups, the chemical information of which is obfuscated. We did not want to remove R groups entirely, since that would remove details valuable for model performance and most R groups are attached to the rest of the molecule through a carbon atom in these KEGG compound entries. Under the same rationale, we replaced repeat structures with ‘C’ (represented with an asterisk ‘*’ symbol in molfiles as compared to ‘R’).

Adding more information to the atom colors enabled us to generate a higher amount of coloring combinations, which further distinguished one metabolite from another, reducing the number of duplicate feature vectors. It is especially important to reduce the number of duplicate feature vectors that map to different labels, since such entries confuse the machine learning model. After the atom colors were generated and the feature vectors of 0-bond-inclusive up to 3-bond-inclusive were concatenated together (Table 1), features corresponding to duplicate columns were dropped since they do not provide additional information and only slow down model training.

#### 2.2.4. Manuel Curation and Filtering

While adding additional details by configuring the atom coloring method (Appendix A) reduced the amount of matching feature vectors mapping to non-matching labels, there were still such entries present. We also noticed the presence of R groups representing proteins or nucleic acids. Such macromolecules cannot be considered metabolites and needed to be removed from the dataset. To facilitate the manual data cleaning, we first programmatically constructed a list of compounds that resulted in duplicate feature vectors. Then, we added to that list compounds with R groups that we detected as potential macromolecules (i.e., nucleic acids or proteins). We determined if an R group was a potential macromolecule by pulling the KEGG entries of only the compounds containing R groups using kegg_pull [17] and searching those entries for keywords in their metadata. The keywords we used were ‘protein’, ‘enzyme’, ‘peptide’, ‘rna’, and ‘dna’. If the entry contained one of these keywords in the NAME or COMMENT field of the KEGG entry, we marked it as a potential protein or nucleic acid and added it to the list.

The list served as a template, which we then manually filled in with instructions to handle each case. Upon inspecting each compound using the KEGG browser, we determined the action to take for each case and recorded that action in the file such that it could be read into the next script in the pipeline and computationally modify the dataset. Some entries were merged into one, meaning one entry was kept and any duplicates removed with the pathway labels unioned into a single set. Other compounds were simply removed and some compounds were retained. Either way, the dataset decreased in size as manually detected invalid entries were handled. The removal of entries resulted in more duplicate columns, necessitating removing duplicate columns once more. The actions taken in the manual dataset cleaning along with the justification for those actions and the amount of data removed for each action are described in Figure 1. After the manual dataset cleaning was complete, the final dataset had 5683 metabolites and 14,656 features (Table 2).

### 2.3. Machine Learning Model Design and Generation

The models trained on the final dataset included RF (popular tree-based method), XGBoost (also tree-based though tends to perform better than the RF while training is slower), and a Multi-Layer Perceptron (MLP) (deep learning method). For the MLP, it was not practical to train it on feature vectors of size 14,656. Pragmatically, deep learning methods require optimized training using a graphics processing unit (GPU), which has GPU memory limitations. Therefore, we trained an autoencoder, a separate deep learning model, to compress the feature vectors. We will refer to these compressed features as the encoded dataset, which contained 10% of the original number of features (Figure 4).

#### 2.3.1. Hyperparameter Tuning

We tuned the hyperparameters of the autoencoder using the Bayesian optimization method [21] and we used the same method for tuning the pathway classifier models. See Table 3 for the list of hyperparameters tuned for each model along with the space of values searched followed by the values that were actually selected. Note that Table 3 shows the range of selected hyperparameter values across training dataset and pathway category combinations (except for the autoencoder, which was not a pathway classifier and only trained on the full dataset). See Appendix A for all the selected hyperparameter values for every combination of training dataset and pathway category. With the hyperparameters decided, we trained the autoencoder and used it to encode the final dataset into the encoded dataset (Figure 4).

#### 2.3.2. Ambiguous and Non-Ambiguous Subset Generation and Training

While some of the compounds had R groups, others had repeating units, i.e., the molfile provided the molecular structure of the initial unit but not that of the repeats. Like the R groups, underlying chemical information of the compounds containing repeating units was obfuscated, making their complete molecular structure unknown. We will refer to such compounds as ambiguous compounds. Hypothesizing that the ambiguous metabolites would have different performance than the non-ambiguous, we created 2 subsets from the full final dataset (Figure 4), i.e., the ambiguous subset (only containing metabolites with either R groups or repeat units) and the non-ambiguous subset (only containing metabolites with neither R groups nor repeat units). The two subsets, therefore, were mutually exclusive with 353 entries in the ambiguous and 5330 in the non-ambiguous (Table 4). For the MLP, we additionally created a corresponding encoded version of the ambiguous and non-ambiguous subsets (Table 4, Figure 4).

We trained a separate binary classifier for each of the 12 pathway labels and on both datasets, i.e., full and non-ambiguous (Figure 4). While the ambiguous subset was used for evaluation, it was too small to train on. Each classifier predicts whether a metabolite is part of the corresponding pathway class or not and the combination of the 12 classifiers determines all the pathway classes a metabolite is a part of. With 12 classifiers trained per dataset, this resulted in 24 different classifiers for each of the 3 models (i.e., RF, XGBoost, and MLP), totaling to 72 different classifiers. We tuned separate hyperparameter sets for each of these 72 classifiers (Figure 4, Table 3) since the best hyperparameters may be dependent on both the dataset trained on and the pathway category being predicted.

#### 2.3.3. Cross-Validation Analysis and Performance Evaluation

We performed a cross-validation (CV) analysis, using a stratified train/test split to maintain the ratio of training entries to test entries in each fold such that the ratio closely matched that of the entire dataset. Use of stratified CV folds has been demonstrated to reduce the variance in model performance across folds, increasing robustness [22]. For each of the 72 classifiers, we trained and evaluated each model over 1000 CV iterations, each iteration consisting of a stratified 95% train/5% test fold split randomly sampled from the dataset. When training on the full dataset, each test fold was divided into 3 test sets, the full fold, the ambiguous entries in the fold, and the non-ambiguous entries in the fold (Figure 4). Evaluation scores were computed for each of the 3 test sets when trained on the full dataset. For the non-ambiguous dataset, all the entries in the fold were, of course, non-ambiguous. So, the two test sets for the non-ambiguous were the entire fold with only non-ambiguous entries and the entire ambiguous subset (the non-ambiguous test set changed with each fold and the ambiguous test set remained the same, being the entirety of the ambiguous entries). After model performance was calculated for the full and non-ambiguous datasets, we did the same for the unfiltered dataset on the XGBoost model only.

Model performance metrics were collected for every combination of model, pathway category, dataset, and test set (Figure 4). The metrics measured include accuracy, precision, recall, F1 score, and Matthews correlation coefficient (MCC) [23]. Since the range of the MCC is between −1 and 1, to make it more comparable to the other metrics, we also included the unit-normalized MCC as described by Cao et al. [24]. For the XGBoost models in particular, we measured the feature importance of each feature in the full dataset. To make the feature importance scores comparable across pathway categories and CV folds, we computed the relative feature importance. The relative feature importance was calculated by dividing the score of each feature in each fold by the maximum feature importance (the score of the most important feature) for a given fold. So, the most important feature for a given fold would have a relative feature importance score of 1.0 and the least important features would have scores of 0.0. For aggregating the score of each feature across CV folds, we decided to use the median since the distribution of the mean minus median differences is fairly wide, suggesting considerable skewness (Appendix A).

### 2.4. Computer Hardware and Coding Details

For the RF hyperparameter tuning and training, we used a set of desktop computers with 64 gigabytes (GB) of random access memory (RAM) and central processing units (CPUs) ranging from 3.4 gigahertz (GHz) to 3.6 GHz, each with 6 hyperthreaded (HT) cores. The CPU chips included ‘Intel(R) Core(TM)i7-2600 CPU@3.40 GHz’, ‘Intel(R) Core(TM) i7-5930K CPU@3.50 GHz’, ‘Intel(R) Core(TM) i7-4930K CPU@3.40 GHz’, ‘Intel(R) Core(TM) i7-6850K CPU@3.60 GHz’. For tuning the hyperparameters of and training the XGBoost and MLP, we used high-performance computing (HPC) machines with up to 187 GB of RAM and ‘Intel^®^ Xeon^®^ Gold 6130 CPU@2.10 GHz’ CPUs. No more than 24 h of compute time was allocated for the XGBoost runs and no more than 72 h for the MLP runs. Both the XGBoost runs and MLP runs had 1 core allocated and used a GPU of up to 12 GB of GPU memory, the name of the GPU card being ‘Tesla P100 PCIe 12 GB’. For the hyperparameter tuning and training of the autoencoder, we used similar HPC machines with up to 187 gigabytes of RAM and ‘Intel^®^ Xeon^®^ Gold 6130 CPU@2.10 GHz’ CPUs, but with ‘Tesla V100 SXM2 32 GB’ cards. Only 1 CPU core and no more than 72 h of compute time were allocated for the autoencoder runs.

All code for this project was written in the Python programming language [25]. The model performance metrics along with the RF model and the stratified CV train/test splitting method were provided with the Sci-kit Learn Python package [26]. The MLP model and autoencoder were created using the Tensorflow deep learning Python package [27]. The XGBoost model and feature importance metric were provided with the XGBoost Python package [28]. Data processing was facilitated using the Numpy [29], Pandas [30], and DuckDB [31] Python packages. Tables and figures were created using the DuckDB [31], Pandas [30], Matplotlib [32], and Seaborn [33] Python packages as well as the Tableau desktop application [34]. The results data (model performance scores, feature importance etc.) for producing the tables and figures in this manuscript were placed in a DuckDB database file, which integrated with Tableau. The same data integrated with Python scripts via SQL queries [35]. All code and data (including the DuckDB database file) for complete reproducibility of the results in this manuscript along with instructions to do so are available in the following Figshare item: 10.6084/m9.figshare.24021480.

## 3. Results

### 3.1. Misclassification Rates

After measuring the misclassification rates for the metabolites in the final dataset (Figure 5b) in the same manner as the unfiltered dataset (Figure 2b and Figure 5a), we see the same pattern as before, where compounds with a higher number of non-hydrogen atoms are more likely to classify correctly. However, the metabolites in the final dataset overall classify correctly more frequently than in the unfiltered dataset (Figure 5).

To emphasize this trend, we see that the average misclassification rates in the sliding window are much lower after filtering metabolites with non-hydrogen atom counts less than 7 (Figure 6). The average misclassification rates are near 0 for the final dataset (Figure 6b,c) as compared to the unfiltered dataset (Figure 3 and Figure 6a). When zooming into the sliding window misclassification rates of the final dataset (Figure 6c), we still see the same pattern of metabolites with higher non-hydrogen atom counts classifying correctly even more frequently, while the misclassification rate across the entire dataset improves (drops) after filtering.

### 3.2. Model Performance

Appendix A contains the model performance scores for all combinations of model trained (i.e., XGBoost, RF, and MLP), dataset trained on (including the unfiltered dataset for the XGBoost model only), test set evaluated on, pathway category predicted, and metric used (i.e., accuracy, precision, recall, F1 score, MCC, and unit-normalized MCC). There are 1152 different combinations, each with four aggregates, i.e., average, standard deviation, median, and maximum.

We can reduce the number of comparisons from 1152 to 96 by taking the overall performance across the pathway categories. One could accomplish this by taking the average and standard deviation of the scores for all 12 pathway categories and each of their 1000 CV folds (i.e., average/standard deviation of 12,000 total scores). However, each pathway category takes up a different proportion of each dataset. Appendix A shows each dataset (i.e., full, non-ambiguous, and unfiltered) and the proportion that each pathway category occupies in the corresponding dataset. Using these proportions as weights, we can calculate the weighted average and standard deviation. Appendix A provides both the unweighted and weighted averages and standard deviations of the scores across all 12 pathway categories.

Since the formulas for the performance metrics involve division, all but accuracy have the possibility of dividing by zero. A division by zero is undefined and therefore invalid, and any CV folds resulting in an invalid result were not included in the aggregation (i.e., average, standard deviation, median, etc.). Appendix A shows the number of valid scores out of the 1000 CV iterations for all 1152 CV analyses. While there were CV analyses with all 1000 folds resulting in a valid score, there were also analyses with some of their folds producing invalid scores for the precision, recall, and F1 score metrics. Appendix A (subset of Appendix A) shows the analyses that had less than 300 valid scores. Notice that all these analyses were evaluations on the ambiguous subset, likely because the ambiguous subset usually produced worse classification and it has less entries, both of which increase the likelihood of a division by zero. However, we see from Appendix A (same as Appendix A except only for the MCC metric) that the MCC scores were valid for all 1000 iterations for every CV analysis. This makes MCC the most reliable metric for the results presented here. While accuracy, of course, also had all valid scores, it can be misleading since a binary classifier that only predicts negatives would score very high on a dataset with unbalanced labels [24]. All pathway categories, except for perhaps ‘Biosynthesis of other secondary metabolites’, are highly unbalanced (Appendix A), favoring negatives. For these reasons, results from now on will be reported for MCC only.

Table 5 provides the weighted averages and standard deviations of the MCC scores for each combination of model, dataset, and test set (same as Appendix A except for only MCC and only provides the weighted averages and standard deviations). We see that the best performing analysis was the one training the XGBoost model on the full training set and full test set. This is surprising considering the overall worse performance of the ambiguous test sets (Table 5), which likely performed worse since they lack the chemical information to completely represent the compound (atoms and bonds are missing in the corresponding molfile). Since the full dataset includes ambiguous metabolites, one would expect it to perform worse than the corresponding non-ambiguous subset. However, we see that the full dataset outperforms the non-ambiguous for every model. Appendix A provides an explanation for this unexpected result, showing the MCC for each pathway category and test set, training XGBoost on the full dataset. As seen in Appendix A, the ambiguous test set outperformed the other two test sets for some pathway categories, especially for ‘Glycan biosynthesis and metabolism’. The ambiguous test set’s higher performance had a greater impact on the full test set than for the pathway categories where the ambiguous performed much worse, resulting in the full test set performing better overall.

To emphasize the final dataset’s improved performance compared to the unfiltered dataset, Table 6 shows the weighted average and standard deviation of the XGBoost’s MCC score for both datasets. The final dataset’s results in Table 6 correspond to the full dataset evaluated on the full test set (Table 5) and the unfiltered dataset’s results correspond to those of the entirety of the unfiltered dataset’s CV folds (Table 5), which did not evaluate different subsets of its folds. These results are consistent with what we observe in Figure 5 and Figure 6. Still, the improvement is only (0.7677 − 0.7454)/0.7454 × 100 ≈ 3%.

Figure 7 displays a violin plot of all the MCC scores for each model including all pathway categories. These scores correspond to the ‘full’ dataset evaluated on the ‘full’ portion of the CV folds. We see from Figure 7 that the XGBoost model performed best overall with the performance of the RF being comparable to that of the MLP. All models experience a wide variance though, with the bulk of CV folds scoring higher, resulting in a skew of the performance across CV folds.

Much of the variance observed in Figure 7 can be attributed to the stark difference in performance across pathway categories, as seen in Figure 8. Using the XGBoost scores only, we see that the ‘Chemical structure and transformation maps’ pathway class is the most difficult to predict given the current data available in KEGG. This may explain why it was left out of past studies involving this machine learning task. Nearly as poor is the performance of ‘Energy metabolism’, with ‘Lipid metabolism’ performing the best. Similar trends occurred for the other two models (Appendix A).

### 3.3. Feature Importance

Figure 9 shows a line plot for each pathway category of the top 50 feature importance scores ordered from the most important feature to the least important feature. Each line plot includes the ordered feature importance for the XGBoost model trained to predict the given pathway category on the full dataset. Each point on the line is a feature where the entire line in each plot represents the top 50, the dependent variable being the median feature importance across CV folds. We see a sharp drop towards a feature importance of 0.0 for most pathway categories, suggesting that only the top few features were particularly important for classification using the XGBoost. Appendix A shows the data used to create Figure 9, with the actual atom colors specified for each of the 50 features for each of the 12 pathway categories, resulting in 600 median feature importance scores.

While there are 600 different scores across the 12 categories (Appendix A), 477 are distinct features, suggesting only sparse overlap, i.e., only a minority of the top 50 features in one pathway category are also some of the top 50 in another category. Appendix A shows an upset plot [36] displaying the amount of overlap of important features across the pathway categories. We see from Appendix A that the largest intersection is the set of features unique to ‘Chemical structure transformation maps’, i.e., the features that are only important for predicting ‘Chemical structure transformation maps’ and are not found among the important features of the other categories. This tells us that most of the top 50 features for predicting ‘Chemical structure transformation maps’ are unique to that category. Beyond that, the largest 12 intersections (Appendix A) are those unique to a given pathway category, further highlighting the sparse overlap of important features between the categories. Since the importance of individual features is highly dependent on the pathway category that they are predicting, a weighted average feature importance across all 12 classes would not be especially meaningful, so we will only show the median feature importance score for each class separately.

Table 7 is a portion of Appendix A, highlighting the top 3 (as compared to top 50) most important features for each pathway category. Both Appendix A and Table 7 additionally provide odds ratios for each pathway label corresponding to the presence of the feature in each entry in the full/final dataset and whether each entry is a positive classification (involved in the pathway class) or a negative classification (excluded from the pathway class). We define an entry as having a feature if the atom color appears at least once in the corresponding metabolite and as not having a feature if the atom color appears zero times in the metabolite. The positive odds in Appendix A and Table 7 are the odds that an entry has the feature given that it is a positive entry and the negative odds are the odds that an entry has the feature given that it is a negative entry. The division of the positive odds by the negative odds results in the odds ratio (Appendix A and Table 7) that indicates whether a feature is important for predicting that a metabolite is involved in the pathway class or whether it is important for predicting exclusion. Values well above one suggest the odds of positive entries having the feature is much higher than the odds of negative entries having the feature, i.e., the feature is associated with pathway involvement. Values well below one suggest the odds of negative entries having the feature is much higher than the odds of positive entries having the feature, i.e., the feature is associated with pathway exclusion. Note that for some features, division by 0 resulted in odds ratios of infinity (∞).

Figure 10 displays molecular structure diagrams of an example metabolite that is involved in each pathway class. The single most important feature for that pathway class is displayed within the metabolite, with the bonds of the atom color being highlighted and the atoms being circled. While the examples in Figure 10 have the atom color occurring once, other metabolites may have the atom color occurring multiple times in the molecule, and the features include the count that a given atom color appears. These diagrams highlight which atom configurations in metabolites are important for predicting their pathway involvement. Some of the most important features branched out to two bonds from the originating atom while others branched out to three and in the case of Xenobiotics biodegradation and metabolism, the 0-bond-inclusive feature of chlorine (the number of chlorine atoms in the compound) was most important.

## 4. Discussion

In this work, we present a new KEGG-based dataset for the machine learning task of predicting the pathway involvement of metabolites. It significantly improves on the KEGG-SMILES dataset used in previous publications on metabolic pathway prediction, which contained duplicate entries and lacked the code and description of its creation. The lack of code precludes updating the dataset as more metabolites are discovered, the duplicate entries invalidate the prior analyses, and the lack of description of its creation makes the dataset suspect in general. Huckvale et al. outlined the optimal requirements of a benchmark dataset for this machine learning task, specifying that it must be reproducible, valid, accessible, and complete [15]. We use the highest standards of computational reproducibility in our dataset [37,38], providing a thoroughly detailed description of how it was created. This includes merging duplicate entries discovered in KEGG, filtering entries with limited chemical information causing inferior classification reliability, and semi-automated manual inspection of a small subset of entries with a range of potential issues (Figure 1). We provide the raw data and code for complete reproducibility when re-generating the dataset, as well as the original scripts for obtaining the raw data in the first place, including instructions for adding to the raw data as KEGG releases updates. Our final dataset with a size of 5683 entries exceeds the size of the de-duplicated KEGG-SMILES dataset of size 4929 by 754 entries. Beyond being reproducible and malleable, the dataset is also valid since it contains no duplicate metabolites and the description of its creation is transparent. And finally, the dataset is complete according to the most up-to-date KEGG data as of 3 July 2023 (with the caveat of filtering entries by non-hydrogen atom count). Finally, this new KEGG-based dataset is maintainable as KEGG changes. We recommend future research in metabolic pathway prediction use our dataset, build off of our dataset, or otherwise use the same standards of scientific computational reproducibility, data validation, accessibility, and completion.

We present strong evidence for the correlation of the number of non-hydrogen atoms in a metabolite and the ability for said metabolite to be classified reliably. This trend is expected, since low information content is always an issue in classification. Considering the clear drop in misclassification rate until reaching a non-hydrogen atom count of seven (Figure 3), we recommend future work using our methods predict on metabolites with at least seven non-hydrogen atoms, as our models were trained on a dataset with metabolites that meet this restriction. This is pragmatic, since the prediction of pathway involvement should primarily focus on molecules where direct pathway involvement is not directly known and the pathway involvement of most molecules with less than seven non-hydrogen atoms is better known. But if predicting on metabolites with less than 7 non-hydrogen atoms is necessary, one will either need to be aware of lower reliability or produce models more capable of predicting on such metabolites. When performing this machine learning task using different models or different datasets, we recommend being cautious of non-hydrogen atom count, monitoring misclassification rates of the metabolites.

We defined ambiguous metabolites as those containing R groups or repeat sequences specified in their molfile such that underlying chemical information is obfuscated. We expected and demonstrated that such metabolites would be more difficult to predict correctly for most pathway categories. However, for some pathway categories, i.e., ‘Glycan biosynthesis and metabolism’ and ‘Lipid metabolism’, ambiguous metabolites surprisingly outperformed the non-ambiguous entries (Appendix A).

By generating features from the atom colors, we make features out of the molecular substructures of the metabolites. Measuring the importance of these features enables biochemists to determine which substructures are associated with the pathway involvement of the corresponding metabolites. Some substructures are associated with a metabolite being present in a pathway category while other substructures indicate that a metabolite is absent from said category (Table 7). The ability to quantify the importance of metabolite substructures and their positive association versus negative association provides insight into what substructures are inclusively or exclusively identifying for a pathway category. For example, the C-C-C-C atom color highlighted in Figure 10 would not be readily thought of as an identifying feature for ‘Glycan biosynthesis and metabolism’; however, this feature helps identify metabolites used in lipopolysaccharide biosynthesis along with the presence of other identifying features.

Next, when comparing the performance of the three machine learning models, XGBoost unsurprisingly performed better overall than the Random Forest model while the MLP deep learning method did not improve on the tree-based methods. This is incongruent with deep learning-based methods exceeding the performance of tree-based methods in past publications (albeit on an invalid dataset). However, those models were more sophisticated than a simple MLP. It could be that such deep learning methods could surpass the performance of the XGBoost trained on our atom color features. Though, the atom color features do provide information on the molecular substructure of the metabolites similar to the graph-based models, albeit in a linearized fashion, and it includes information not just on atom configuration but also stereochemistry and bond order. It is still an open question whether models capable of processing more complex data structures can improve upon the performance of XGBoost trained on a tabular dataset. And to our knowledge, such models have yet to incorporate additional information beyond simple backbone molecular structure, such as atom stereochemistry, bond stereochemistry, and bond order.

We recommend using separate classifiers per pathway category. Depending on the pathway category that a classifier is being trained to predict, different hyperparameter values will result from the hyperparameter tuning. We also see that the importance of the features used is highly dependent on the pathway category being predicted (Appendix A) while the majority of features have little to no importance (Figure 9). If future work uses our atom coloring method to generate features, one may consider selecting features based on importance. However, one should be mindful of the pathway class being predicted since different target classes will require different features selected. It is possible that the important features will change further if training models are to predict more specific pathway classes, and we recommend using separate binary classifiers for the more specific pathway classes as well and perhaps a hierarchical classification method.

While the weighted average MCC of XGBoost trained on our final dataset (full feature set, full test set) was 0.7677 with a weighted standard deviation of 0.1540 (Table 6), these weighted aggregates include ‘Chemical structure transformation maps’, the worst performing pathway category (Figure 8). This category was excluded from previous publications on this machine learning task, including the most recent model for metabolic pathway prediction proposed by Du et al. called the MLGL-MP [14]. Huckvale et al. re-ran the MLGL-MP on a de-duplicated version of the KEGG-SMILES dataset [15], making it more comparable to our own dataset (though even the de-duplicated version is suspect). Table 8 shows that the MCC improves significantly when ‘Chemical structure transformation maps’ are removed from the weighted average and weighted standard deviation calculations. We also see from Table 8 that the F1 score of XGBoost trained on our dataset is comparable to that of the MLGL-MP trained on the de-duplicated version of the KEGG-SMILES dataset, keeping in mind that theirs was not a weighted average since their model predicted all the pathway categories at once rather than separating into an isolated classifier per pathway class. It should also be noted that the MLGL-MP was originally evaluated using the test set in each training epoch and choosing the highest scores from multiple evaluations, thus using the test set for model selection [15], while we instead followed the best practice of training the models completely and evaluating on the test set only once per CV fold. We do not compare to the standard deviation of the MLGL-MP, since the MLGL-MP was only evaluated on 10 unique folds, which does not provide a reasonable estimate of the actual model performance variation.

In all of these cross-validation analyses and performance evaluations, KEGG is treated as a gold standard, making the assumption that KEGG’s description of pathway involvement is complete and without error. This assumption is necessary for training and evaluation and should be somewhat reasonable for central metabolism; however, this assumption is clearly not true, since KEGG is growing. Thus, there are implications with using a “gold standard” that is evolving. Besides improved machine learning methods and models, metabolic pathway prediction may also improve with more positive entries, i.e., metabolites that are involved in particular pathway classes (positive) as compared to not being involved in said pathway class (negative). A higher amount of positive entries was simulated by duplicating already extant positive entries in the SMILES dataset as shown by Huckvale et al. [15], which of course resulted in impressive scores for this machine learning task in past publications but rendered the dataset invalid. However, the higher scores from duplicated entries do provide evidence that having more non-duplicate (real/valid) positive entries can greatly improve model performance, including in currently poor performing categories like ‘Energy metabolism’. For low performing pathways, more positive entries may be added to KEGG over time. However, additional positive metabolites may already be available in other data sources such as MetaCyc and PubChem.

Since the overall performance as well as the variance in performance are greatly dependent on the pathway category being predicted, certain use-cases may need to exclude certain pathway categories. As illustrated with the violin plots of MCC scores per pathway category in Figure 8 as well as Appendix A, ‘Chemical structure transformation maps’, Energy metabolism‘, and ‘Metabolism of other amino acids’ pathway category predictions fall below a median MCC performance of 0.6 and should likely be excluded from many practical applications.

## 5. Conclusions

We present a new KEGG-based benchmark dataset for the machine learning task of metabolic pathway prediction, which is valid, comprehensive, completely transparent, fully reproducible, readily accessible via our Figshare, and maintainable as KEGG changes. In our hands, the XGBoost machine learning method outperformed both Random Forest and MLP with autoencoder methods for the classification of metabolites to 12 KEGG-defined pathway categories. While the scores attained with the XGBoost model trained on our dataset are seemingly less impressive than those obtained with other methods developed on the KEGG-SMILES dataset, we maintain that the previous publications are invalidated with the duplicate entries in the KEGG-SMILES dataset. Therefore, the results of KEGG metabolic pathway prediction performance presented here are trustworthy. Furthermore, the atom color features employed in our methods provide chemical insight into which molecular substructures are informative for pathway category prediction. Finally, we recommend that individual pathway category prediction performance be evaluated for each potential use case and application.

## Figures and Tables

**Figure 1 metabolites-13-01120-f001:**
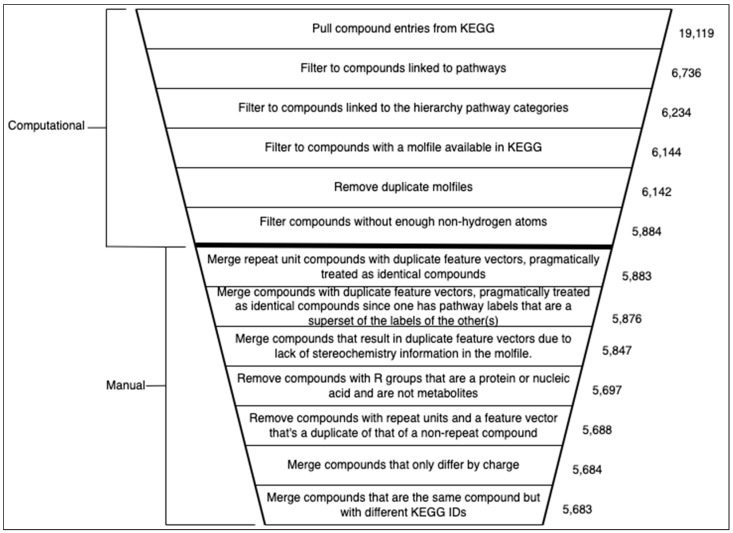
Diagram summarizing both the computational and manual trimming of the metabolite entries in the dataset, the entries being based on compounds from the KEGG database. Each step in the data cleaning process resulted in less entries in the dataset until arriving at the final dataset. The computational steps were entirely automated with Python scripts while the manual portion was semi-automated. The manual portion included filling out a file with a text editor containing a list of KEGG compound entries from the dataset after filtering by non-hydrogen atom count. We filled in the file with instructions on how to handle the compounds based on manual inspection of their properties. The file was initially generated as a template using a script and an additional script used the instructions in the file after we manually added them to it. Whether merging or completely removing manually inspected metabolites, the dataset size continued to decrease with the manual dataset cleaning.

**Figure 2 metabolites-13-01120-f002:**
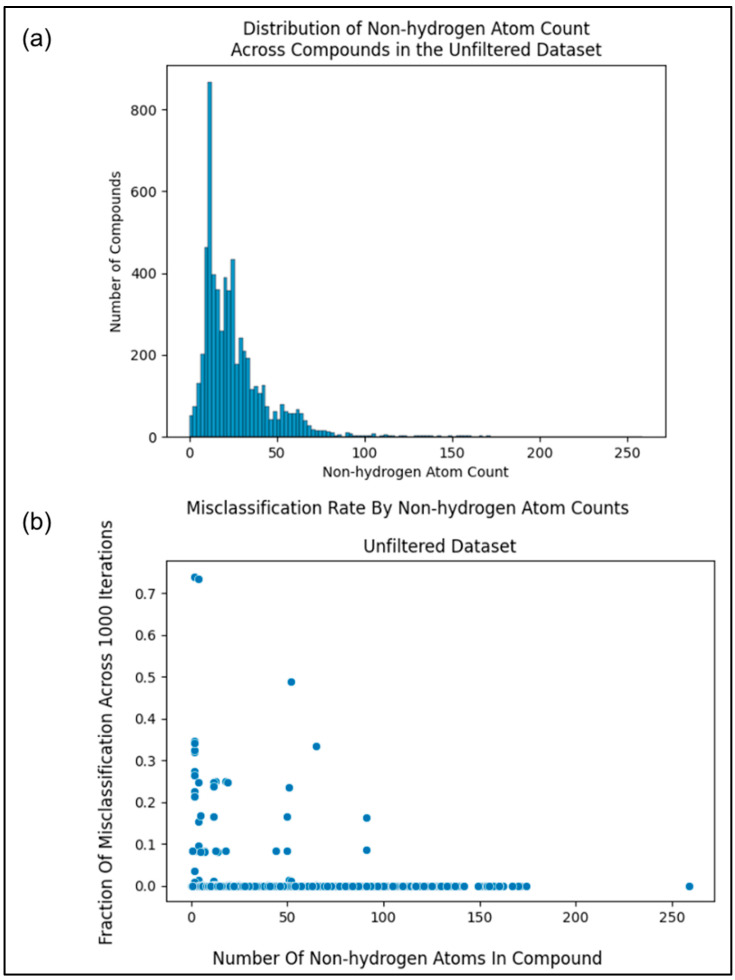
(**a**) Histogram showing the distribution of the number of non-hydrogen atoms within the metabolites in the unfiltered (initial) dataset. (**b**) Scatterplot comparing the misclassification rate of metabolites in the unfiltered (initial) dataset to the non-hydrogen atom count of said metabolites.

**Figure 3 metabolites-13-01120-f003:**
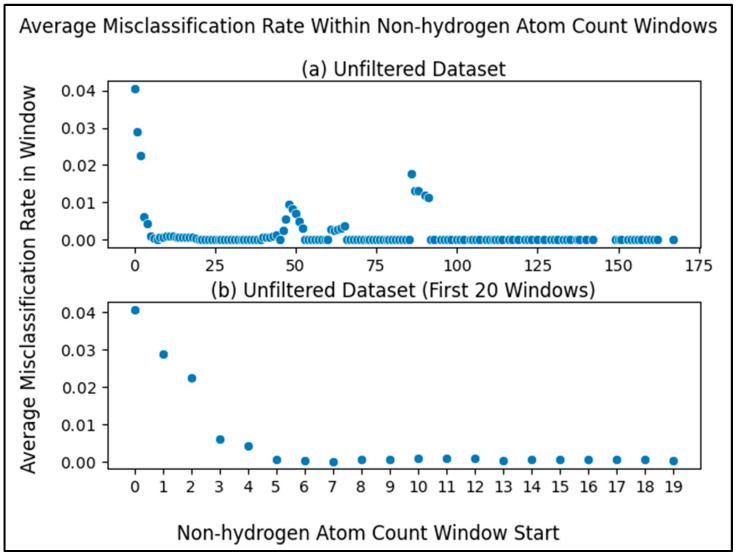
Average misclassification rate of metabolites within a sliding window, where the metabolites within each window are those with a non-hydrogen atom within the range of the window. The x-axis portion of each datapoint represents the start of each window where the window size is 5, e.g., the first datapoint ranges from a non-hydrogen atom count of 0 to 4. (**a**) Sliding window scatterplot of the metabolites in the unfiltered (initial) dataset. (**b**) Same as Figure 3a but zoomed into the first 20 points (0 to 19) to more clearly see the first local minimum.

**Figure 4 metabolites-13-01120-f004:**
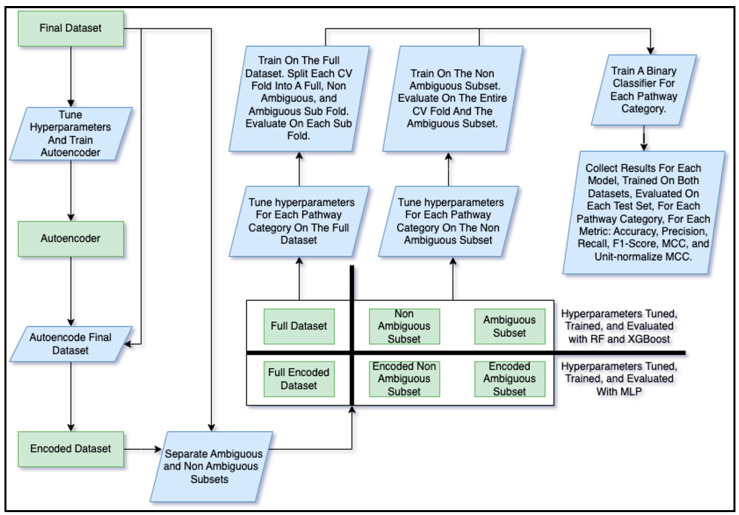
Flowchart of the training and model evaluation pipeline. The autoencoder was created to convert the final dataset into an encoded version with less features (feature reduction) such that it could more practically be used to train the MLP. The autoencoder had hyperparameters tuned prior to the final training as did the three pathway classifier models. Both the non-encoded and encoded datasets were divided into three subsets, i.e., the ambiguous (entries with R-groups or repeating groups), non-ambiguous, and full. The MLP was trained on both the full and non-ambiguous encoded datasets and the RF and XGBoost models were trained on the non-encoded counterparts. When trained on the full dataset, models were evaluated on the full, non-ambiguous, and ambiguous test sets derived from each CV fold. When trained on the non-ambiguous dataset, models were evaluated on the entire non-ambiguous fold and the entire ambiguous subset. A separate binary classifier was trained for each combination of pathway category, dataset, and model. Various evaluation metrics were computed for each test set (full, non-ambiguous, and ambiguous test sets when trained on the full dataset and non-ambiguous and ambiguous test sets when trained on the non-ambiguous dataset) that was evaluated using each classifier.

**Figure 5 metabolites-13-01120-f005:**
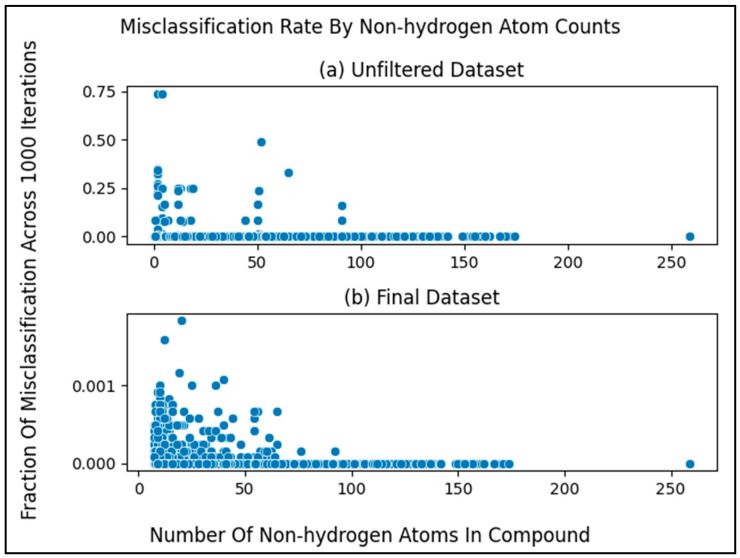
Misclassification rates of metabolites compared to the number of non-hydrogen atoms within them. (**a**) Same as Figure 2b but copied over for comparison. (**b**) Same as Figure 5a but for the final dataset. We observe greatly improved (lower) misclassification rates upon filtering metabolites by non-hydrogen atom count. Notice the drastically different y-axes scales between (**a**,**b**).

**Figure 6 metabolites-13-01120-f006:**
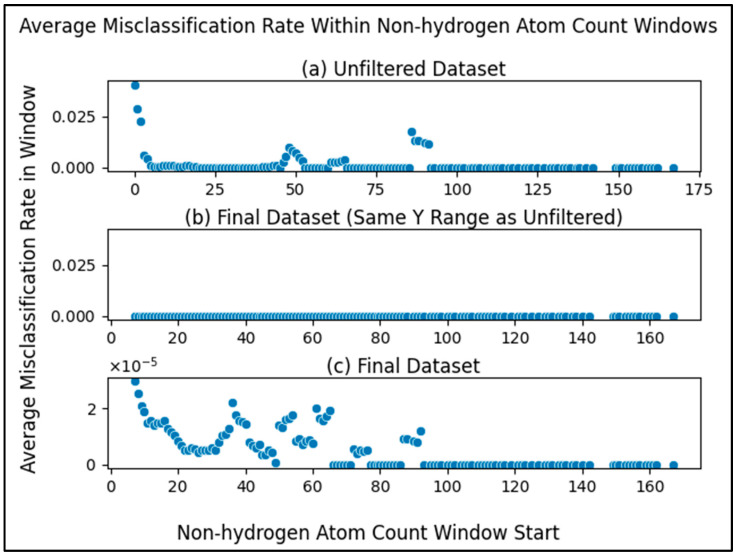
Average misclassification rate of metabolites within windows (ranges) of non-hydrogen atom count for both the unfiltered and the final datasets; (**a**) same as Figure 3a with y-axis rescaled for comparison; (**b**) same as Figure 6a but for the final dataset rather than the unfiltered dataset; (**c**) same as Figure 6b but zoomed in by fitting the y-axis to the data to be able to compare the individual datapoints. We observe that average misclassification of metabolites within windows of non-hydrogen atom count improves (drops) after filtering while still maintaining the trend of metabolites with more non-hydrogen atoms having even lower misclassification rates. Notice the drastically lower y-axis scale of (**c**).

**Figure 7 metabolites-13-01120-f007:**
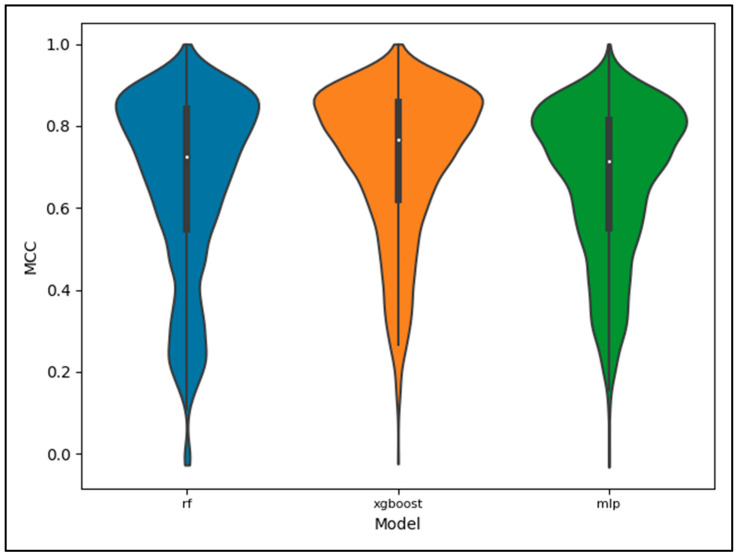
Violin plot displaying the distribution of scores for the MCC metric, full dataset, and full test set by model. The distributions include scores for all pathway categories, with lower performing pathway categories having the bulk of their scores occupy the lower end of the distributions and the higher performing pathway categories occupying the higher end of the distributions.

**Figure 8 metabolites-13-01120-f008:**
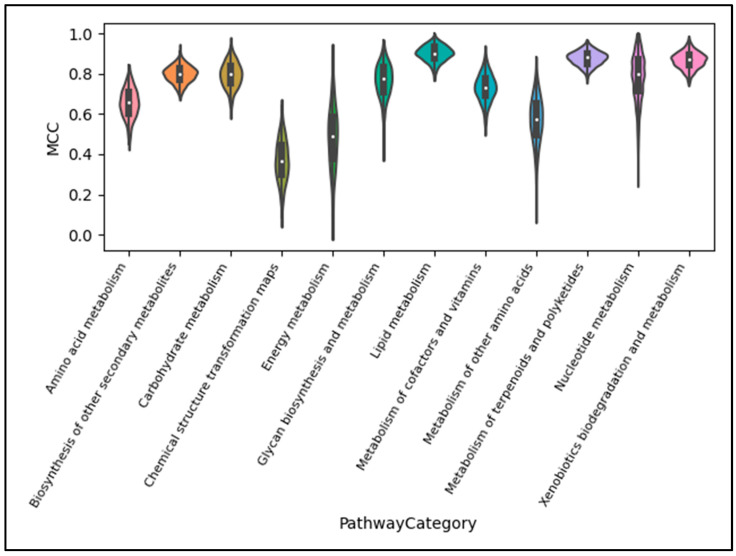
Violin plot displaying the distribution of scores for the MCC metric, XGBoost model, full dataset, and full test set by pathway category. We see that the median performance and variance of performance across CV folds greatly depend on not just the model used, as seen in Figure 7, but also the pathway category being predicted.

**Figure 9 metabolites-13-01120-f009:**
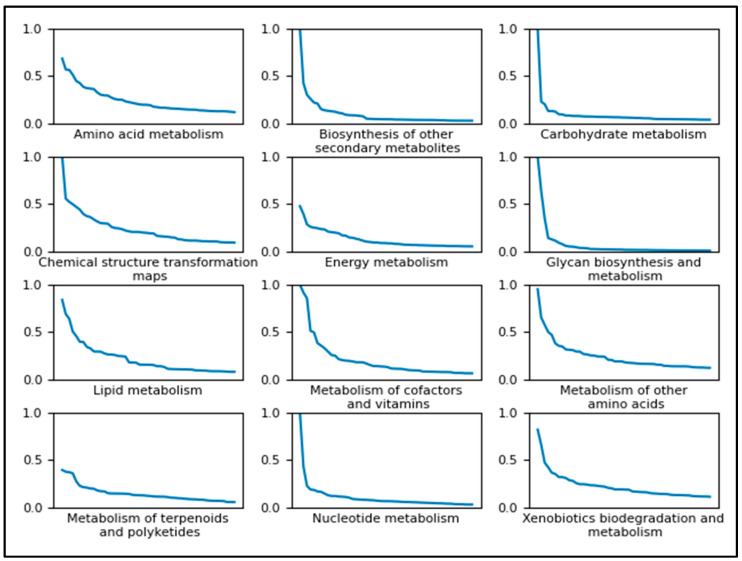
Line plots of the top 50 most important features for predicting each pathway category using the XGBoost model trained on the full dataset. The independent variable (x-axis) of each plot is each discrete feature ordered by its importance along the x-axis and the dependent variable (y-axis) is the relative importance score of the given feature. Most pathway categories experience a drop close to 0 within the top 50 features.

**Figure 10 metabolites-13-01120-f010:**
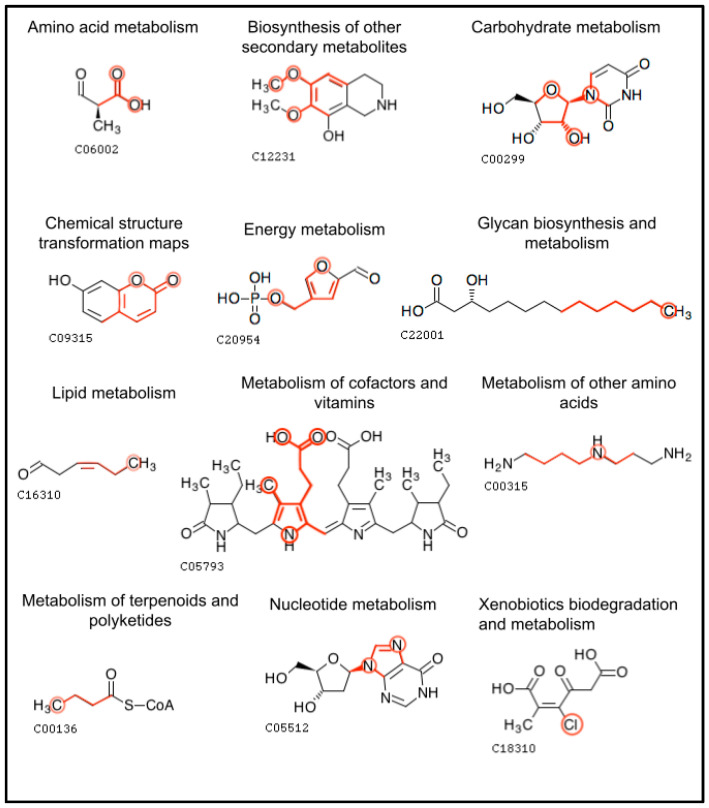
Example metabolites within each pathway category that contain the single most important feature for that pathway category. The bonds of the feature are highlighted with the atoms in the feature circled (except for the carbon atoms that are part of carbon–carbon chains). Atom colors of various bond inclusivity (e.g., 0-bond-inclusive, 2-bond-inclusive, etc.) were most important.

**Table 1 metabolites-13-01120-t001:** Number of features in concatenated feature sets by atom color bond inclusivity.

Atom Color Bond Inclusivity(s)	Number of Features
0 Bond (Atom Count Only)	23
0 and 1 Bond Concatenated	1100
0, 1, and 2 Bond Concatenated	8625
0, 1, 2, and 3 Bond Concatenated	14,923

Atom color features generated with higher bond inclusivity result in more possible combinations of atom colors and therefore more features. Concatenating to features generated from lower bond inclusivity results in even more features.

**Table 2 metabolites-13-01120-t002:** Number of entries and features before and after the manual dataset cleaning.

Stage	Number of Entries	Number of Features
Before Manual Dataset Cleaning	5884	14,923
After Manual Dataset Cleaning	5683	14,656

The number of entries decreased because of the removal and merging of entries in the manual dataset cleaning. Decreasing the entries resulted in duplicate columns once again, the de-duplication of which resulted in another decrease in features.

**Table 3 metabolites-13-01120-t003:** The hyperparameters tuned for each model.

Model	Hyperparameter Name	Hyperparameter Values Searched	Hyperparameter Value(s) Selected
XGBoost	Alpha	0.0–2.0	0.0–2.0
	Booster	dart, gbtree	dart, gbtree
	ETA	0.01–0.5	0.0812–0.5
	Lambda	0.0–2.0	0.0–2.0
	Maximum Depth	6–9	6–9
	Minimum Split Loss	0.0–2.0	0.0–0.9333
	Scale Position Weight	0.5–2.0	0.9617–2.0
	Subsample	0.2–1.0	0.6159–1.0
RF	CCP Alpha	0.0–0.9	0.0–0.0047
	Class Weight	balanced, balanced_subsample	balanced, balanced_subsample
	Criterion	entropy, gini, log_loss	entropy, gini, log_loss
MLP	Activation	relu, selu	relu, selu
	Beta 1	0.0001–0.99999	0.0001–0.9662
	Beta 2	0.2–0.99999	0.3070–0.99999
	Bias Regularization Factor	0.0–0.01	0.0–0.01
	Bias Regularization Type	l1, l2, l1l2	l1, l2, l1l2
	Epsilon	0.00000001–0.0001	0.00000001–0.0001
	Jitter	0.00001–0.05	0.00001–0.05
	Kernel Regularization Factor	0.0–0.005	0.0–0.005
	Kernel Regularization Type	l1, l2	l1, l2
	Learning Rate	0.0000001–0.001	0.0000167–0.001
	Negative Class Weight	0.2–1.0	0.2–1.0
	Number of Layers	2–5	2–5
	Positive Class Weight	1.0–5.0	1.0–5.0
	Train Set Augmentation Size	0.001–1.0	0.001–1.0
Autoencoder	Activation	elu, relu, selu	selu
	Learning Rate	0.00001–0.001	0.00002176
	Number of Layers	4–6	6

Every model had hyperparameters that were tuned including the autoencoder for making the encoded datasets, the MLP trained on those encoded datasets, and the other two pathway classifier models trained on the non-encoded datasets. The hyperparameters acted as configuration for the models in general as well as configuration for their respective training algorithms. The space of hyperparameters searched is provided for numeric hyperparameters as a range of the lowest to highest number while for categorical hyperparameters, the set of categories. The hyperparameter value(s) selected are provided using the range of the lowest number selected to the highest (numeric) or the subset of categories selected (categorical) across all combinations of pathway category and dataset for the XGBoost, RF, and MLP models. Since the autoencoder did not train on different pathway categories nor different datasets (it was only trained on the full dataset and the features themselves were the labels, so different pathway category combinations are not applicable for the autoencoder), the single selected values are provided rather than a subset or numeric range.

**Table 4 metabolites-13-01120-t004:** Number of entries and features in each subset.

Dataset	Number of Entries	Number of Features
Full Dataset	5683	14,656
Ambiguous Subset	353	14,656
Non-ambiguous Subset	5330	14,656
Full Encoded	5683	1465
Ambiguous Encoded	353	1465
Non-ambiguous Encoded	5330	1465

The full dataset, including both the ambiguous and non-ambiguous subsets, had a size equal to the sum of their sizes. The respective encoded versions had the same number of entries. However, there was only 10% of the number of features in the encoded version to make training and evaluating the MLP more practical.

**Table 5 metabolites-13-01120-t005:** Weighted average MCC.

Model	Dataset	Test Set	Weighted Average	Weighted Standard Deviation
XGBoost	Full	Ambiguous	0.4543	0.4567
		Full	**0.7677**	0.1540
		Non-ambiguous	0.7644	0.1552
	Non-ambiguous	Ambiguous	0.2435	0.1257
		Non-ambiguous	0.7606	0.1569
	Unfiltered	Unfiltered	0.7454	0.1511
Random Forest	Full	Ambiguous	0.4101	0.4483
		Full	0.7361	0.1874
		Non-ambiguous	0.7329	0.1890
	Non-ambiguous	Ambiguous	0.2179	**0.1147**
		Non-ambiguous	0.7372	0.1751
Multi-layer Perceptron	Full	Ambiguous	0.4225	0.4409
		Full	0.7240	0.1615
		Non-ambiguous	0.7201	0.1634
	Non-ambiguous	Ambiguous	0.2721	0.1553
		Non-ambiguous	0.7136	0.1617

Using the pathway category proportions of each dataset as weights, we calculated the weighted average and standard deviation of the MCC scores across all pathway categories and CV folds (12,000 total scores per aggregation) for each dataset and test set. The highest average and lowest standard deviation are in bold.

**Table 6 metabolites-13-01120-t006:** Weighted Average XGBoost MCC of the final dataset compared to the unfiltered dataset.

Dataset	Weighted Average	Weighted Standard Deviation
Final	0.7677	0.1540
Unfiltered	0.7454	0.1511

Same weighted average MCC scores as seen in Table 5, but only for the XGBoost model trained on the final dataset and the unfiltered dataset, using the entirety of the CV folds for each (the ‘full’ test set for the ‘full’ dataset while the ‘unfiltered’ dataset was not divided into sub-folds i.e., it only had the one ‘unfiltered’ test set).

**Table 7 metabolites-13-01120-t007:** The top 3 most important features with their median feature importance score and odds ratio corresponding to the presence of the feature in dataset entries and the positivity of the classification of said entries.

Pathway Category	Feature Name	Median Feature Importance	Positive Odds	Negative Odds	Odds Ratio
Amino acid metabolism	O0(O0((C0.10)))(C0((C2.10))((O0.10))((O0.20)))	0.685652852	0.055646483	0.007492113	7.427340984
	O0(O0((C0.10)))(C0((C1.10))((O0.10))((O0.20)))	0.570630789	0.112929620	0.021687698	5.207082272
	C0(C0((C0.10))((N0.10))((O0.20)))(C0((C0.10)))(N0(2_(C0.10)))(O0((C0.20)))	0.566060364	0.019639935	0.002957413	6.640916348
Biosynthesis of other secondary metabolites	C0(C0((C0.10))((C0.21))((O0.10)))(C0((C0.10))((C0.21)))(C0((C0.10))((C0.21))((O0.10)))(O0(2_(C0.10)))	1	0.061911169	0.002144389	28.871242523
	C0(C0((C0.10))((C0.21)))(C0((C0.10))((C0.21)))(C0((C0.10))((C0.21))((N0.10)))	0.425127536	0.096904442	0.004765308	20.335397720
	C0(C0((C0.10))((C0.21)))(C0((C0.10))((C2.10)))(C0((C0.21))((C1.10))((C2.10)))	0.303291023	0.011440108	0	∞
Carbohydrate metabolism	C2(C2((C1.10))((N0.11))((O0.10)))(C1((C1.10))((C2.10))((O0.16)))(N0(2_(C0.10))((C2.11)))(O0((C1.10))((C2.10)))	1	0.132295713	0.020506868	6.451288223
	C0(C0((C1.10))((N0.10))((O0.10)))	0.230698660	0.019455252	0	∞
	O0(O0((C0.10))((P0.10)))	0.204007506	0.410505831	0.154575348	2.655700445
Chemical structure transformation maps	C0(C0((C0.10))((C0.21)))(C0((C0.10))((C0.21)))(C0((C0.10))((O0.10))((O0.20)))(C0(2_(C0.10))((C0.21)))(O0((C0.20)))(O0(2_(C0.10)))	1	0.013729977	0.001143728	12.004576683
	N0(N0(3_(C0.10)))(C0((C0.10))((N0.10)))(C0((C0.10))((N0.10))((O0.20)))(C0(2_(C0.10))((N0.10)))	0.558694124	0.009153318	0.002287457	4.001525402
	C0(C0((C2.16)))(C2(2_(C0.10))((C0.16)))	0.523999751	0.020594966	0.007434235	2.770287037
Energy metabolism	C0(C0(2_(C0.10))((C0.2-1)))(C0((C0.10))((C0.21)))(C0((C0.10))((O0.10)))(C0((C0.2-1))((O0.10)))	0.478080899	0.034682080	0	∞
	C0(C0((C1.11))((O0.10)))(C1((C0.11))((C1.10))((O0.10)))(O0((C0.10))((P0.10)))(C1(2_(C1.10))((O0.16)))(O0(2_(C1.10)))(P0(3_(O0.10))((O0.20)))	0.395558566	0.063583814	0.002722323	23.356454849
	C0(C0(2_(C0.10)))(2_C0(2_(C0.10)))(C0((C0.10))((C1.10)))(C0((C0.10))((O0.10))((O0.20)))	0.286503404	0.017341040	0	∞
Glycan biosynthesis and metabolism	C0(C0(2_(C0.10)))(2_C0(2_(C0.10)))(C0((C0.10)))(C0(2_(C0.10)))	1	0.268518507	0.038813211	6.918224812
	O0(O0((C0.10))((P0.10)))(C0((C1.11))((O0.10)))(P0(3_(O0.10))((O0.20)))(C1((C0.11))((C1.10))((O0.10)))(O0((P0.10)))(O0((P0.20)))(O0(2_(P0.10)))	0.639740050	0.287037045	0.026870685	10.682163239
	C0(C0((C0.10)))(C0((C0.10))((N0.10))((O0.20)))	0.348376751	0.444444448	0.016980780	26.173381805
Lipid metabolism	C0(C0(2_(C0.10)))(C0((C0.10)))(C0((C0.10))((C0.21)))	0.840231776	0.088626295	0	∞
	H0	0.691794932	0.330871493	0.163004398	2.029831648
	C0(C0((C0.10))((C0.21)))(C0((C0.10))((C0.21)))(C0(2_(C0.10)))	0.640349686	0.271787286	0.003995206	68.028358459
Metabolism of cofactors and vitamins	C0(C0(2_(C0.10)))(C0(2_(C0.10)))(C0(2_(C0.10))((C0.21)))(C0((C0.10))((C0.21))((N0.10)))(C0((C0.10))((O0.10))((O0.20)))(C0(2_(C0.10))((C0.21)))	1	0.085610203	0	∞
	C0(C0((C0.10))((C0.2-1)))(C0(2_(C0.10)))(C0(2_(C0.10))((C0.2-1)))(C0((C0.10)))(C0(2_(C0.10)))(C0(2_(C0.10))((C0.2-1)))	0.917271197	0.018214935	0.000389560	46.757740021
	C0(C0((C0.10))((C0.2-1))((O0.10)))(C0(2_(C0.10))((C0.2-1)))(C0(2_(C0.10))((C0.21)))(O0((C0.10))((C1.10)))	0.857100129	0.014571949	0	∞
Metabolism of other amino acids	C0(C0((C0.10))((N0.10)))(C0(2_(C0.10)))(N0(2_(C0.10)))(C0((C0.10))((N0.10)))(C0(2_(C0.10)))	0.953770041	0.043795619	0.001294139	33.841503143
	C0(C0((C0.10))((P0.10)))	0.654412329	0.124087594	0.000554631	223.729919434
	C0(C0((C0.10))((N0.10)))(C0(2_(C0.10)))(N0((C0.10)))(C0((C0.10))((N0.10))((O0.20)))	0.576080739	0.014598540	0	∞
Metabolism of terpenoids and polyketides	C0(C0(2_(C0.10)))(C0((C0.10)))(C0(2_(C0.10)))	0.395134032	0.004574565	0.074509807	0.061395485
	C0(C0((C1.10))((C2.10)))	0.377435565	0.156450137	0.036819171	4.249148846
	C0(C0(2_(C0.10))((C0.21)))(C0((C0.10))((C0.21))((O0.10)))(C0(2_(C0.10))((C0.2-1)))(C0(2_(C0.10))((O0.20)))	0.372357339	0.043000914	0	∞
Nucleotide metabolism	C0(C0((N0.10))((N0.21)))(N0((C0.10))((C0.21)))(N0(2_(C0.10))((C2.11)))	1	0.313609481	0.014508524	21.615531921
	O0(O0((C1.16))((P0.10)))(C1(2_(C1.10))((O0.16)))(P0(3_(O0.10))((O0.20)))	0.429942638	0.082840234	0.000906783	91.356216431
	N0(N0(2_(C0.10))((C2.11)))	0.228152484	0.473372787	0.044613712	10.610477448
Xenobiotics biodegradation and metabolism	Cl0	0.821942925	0.160809368	0.007166948	22.437637329
	C0(C0((C0.10))((C0.21)))(C0((C0.10))((C0.21)))(C0(2_(C0.10))((C0.21)))	0.662525833	0.246006384	0.073566608	3.343995094
	C0(C0(2_(C0.10)))(C0(2_(C0.10)))(C0(2_(C0.10))((O0.20)))(2_C0(2_(C0.10))((C0.21)))(O0((C0.20)))	0.472949028	0.012779552	0.000421585	30.313098907

Table 7 only differs from Appendix A in that it highlights the top 3 most important features as compared to the top 50 used to generate Figure 9. Each feature is indicated as most important for predicting the respective pathway class with its median relative feature importance across all 1000 CV folds, training XGBoost on the full/final dataset. The feature name is a string representation of the atom color, i.e., the local chemical neighborhood or subgraph around a specific atom. A dataset entry having a feature is defined as the atom color being present in the molecular structure of the corresponding metabolite at least once. A positive entry is involved in the corresponding pathway class and a negative entry is excluded from the class (i.e., having a positive or negative label in binary classification). The positive odds are the odds that an entry has a feature given that it is positive and the negative odds are the odds that an entry has a feature given that it is negative. The quotient of the two values results in the odds ratio indicating whether a feature is important for predicting inclusion in a pathway class or exclusion, with high values indicating the feature is associated with pathway involvement and low values near or equal to 0 indicating the feature is associated with a lack of pathway involvement. Digits following the elemental identity of atoms (e.g., C, O, Cl, etc.) indicate the chirality of those atoms, i.e., a 0 means it is not a chiral atom and above 0 indicates it is chiral. Bear in mind that the atom coloring strings were designed for uniqueness and not human readability.

**Table 8 metabolites-13-01120-t008:** Performance of the XGBoost trained on our dataset compared to that of the MLGL-MP trained on the SMILES dataset.

**Model/Dataset**	**XGBoost Trained/Tested on Our Dataset**		**MLGL-MP Trained/Tested on The SMILES Dataset**
**Metric**	**Weighted Average**	**Weighted Standard Deviation**	**Average**
**F1 Score**	0.8180	0.1190	0.8224
**MCC**	0.7933	0.1196	N/A

The weighted average and weighted standard deviations of the MCC and F1 score of the XGBoost after excluding ‘Chemical structure transformation maps’, the worst performing pathway category, which was excluded from previous publications on the metabolic pathway prediction machine learning task. The most recent model proposed for metabolic pathway prediction, i.e., the MLGL-MP deep learning model, scored a slightly higher F1 score when trained on the de-duplicated version of the SMILES dataset compared to the weighted average of XGBoost on our new dataset. It should be noted that the MLGL-MP predicted all possible 11 labels at once rather than calculating the weighted average scores of individual binary classifiers. The standard deviation of the MLGL-MP’s F1 score is not compared since it was evaluated on only 10 unique folds which does not provide a good estimate on performance variance.

## Data Availability

The data and code for complete reproducibility of the results in this work are available via FigShare at: https://doi.org/10.6084/m9.figshare.24021480.

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
