# Peer review of "Benchmark Dataset for Training Machine Learning Models to Predict the Pathway Involvement of Metabolites"

_metabolites, 2023, doi:10.3390/metabo13111120_

Round 1

Reviewer 1 Report

Comments and Suggestions for Authors

This paper presents a study that aimed to 1) build a high-quality KEGG-based dataset of metabolite involved pathways and 2) classify metabolites into different pathway categories using machine learning models. The researchers trained and evaluated three models (Random Forest, Multi-layer Perceptron, and XGBoost) using cross-validation analysis and measured various performance metrics. The findings demonstrate the potential of machine learning models in predicting pathway involvement of metabolites, providing valuable insights for metabolomics research, and aiding in the understanding of metabolic pathways. The new high-quality dataset is important for the field to developing better models. The manuscript is very detailed, and I would consider making it more structured, such as by adding sub-titles for the Methods section and adding more descriptive sub-titles for the results section.

1.     Some of the key terms are not defined, for example the misclassification rate and ambiguous subset. On Figure 3A, the average misclassification rate of 4% seems to be low, why would you exclude your model’s ability to predict those compounds with 6 or less non-hydrogen atoms.

2.     It is not clear why non-hydrogen atom count becomes the focus of the study. A detailed background introduction for the concept should be added.

3.     The study primarily relied on cross-validation analysis for model evaluation. Is it possible to include external validations?

4.     The Ambiguous dataset is very small (n=353), do you think this drives the performance reported on Table 5. What is the biological meaning of using ambiguous and non-ambiguous sets separately?

5.     What are the feature names for Table 7?

Comments on the Quality of English Language

Some typo needs to be fixed. For example in line 562 'now' should be 'new'.

Author Response

Reviewer 1:

This paper presents a study that aimed to 1) build a high-quality KEGG-based dataset of metabolite involved pathways and 2) classify metabolites into different pathway categories using machine learning models. The researchers trained and evaluated three models (Random Forest, Multi-layer Perceptron, and XGBoost) using cross-validation analysis and measured various performance metrics. The findings demonstrate the potential of machine learning models in predicting pathway involvement of metabolites, providing valuable insights for metabolomics research, and aiding in the understanding of metabolic pathways. The new high-quality dataset is important for the field to developing better models. The manuscript is very detailed, and I would consider making it more structured, such as by adding sub-titles for the Methods section and adding more descriptive sub-titles for the results section.

Response:

We thank the reviewer for their positive comments.  We have added sub-sections/titles to the Methods section as suggested. The Results section already had sub-sections with reasonable sub-titles. So, we do not see think more should be added.

Issue 1:

  1. Some of the key terms are not defined, for example the misclassification rate and ambiguous subset. On Figure 3A, the average misclassification rate of 4% seems to be low, why would you exclude your model’s ability to predict those compounds with 6 or less non-hydrogen atoms.

Response:

This is a philosophical point.  We are trying to minimize the effects of low information content on machine learning. We removed just 4% of the unfiltered dataset to minimize these effects (see Figure 1).  These results also identify information content limitations of machine learning and thus provide boundary conditions for optimal machine learning performance.   Also, pragmatically, any machine learning method for pathway prediction will not be used by itself.  Molecules whose pathway involvement are known, i.e. are identified in KEGG, and thus do not need to be classified by a machine learning method and should not be.  Most of the very small biomolecules are well documented and annotated in KEGG.  So, we are primarily developing a dataset that can train machine learning methods that can be applied to molecules where pathway involvement is not directly known.  We have added the following statements to makes these points clearer:

“This is pragmatic, since the prediction of pathway involvement should primarily focus on molecules where direct pathway involvement is not directly known and the pathway involvement of most molecules with less than 7 non-hydrogen atoms is better known.”

Issue 2:

  1. It is not clear why non-hydrogen atom count becomes the focus of the study. A detailed background introduction for the concept should be added.

Response:

Non-hydrogens atom count is NOT the focus of the study.  But we wanted to be thorough with describing how the benchmark dataset was constructed.  But information content is always an issue with machine learning and statistical methods.  We have added the following as some additional background information on the topic, but we do not want to emphasize the concept any further:

“We present strong evidence for the correlation of the number of non-hydrogen atoms in a metabolite and the ability for said metabolite to be classified reliably.  This trend is expected, since low information content is always an issue in classification.”

Issue 3:

  1. The study primarily relied on cross-validation analysis for model evaluation. Is it possible to include external validations?

Response:

 We are taking the pathway involvement annotations from KEGG as a gold standard.  From this perspective, the cross-validation is an external validation.  However, treating KEGG as a gold standard assumes that it is complete, which no metabolic dataset is.  We are still discovering new metabolic reactions; however, most of central metabolism is thought to be fairly complete.  We have added the following statements about treating KEGG as a gold standard in terms of the cross-validation and the assumptions that entails:

“In all of these cross-validation analyses and performance evaluations, KEGG is treated as a gold standard, making the assumption that KEGG’s description of pathway involvement is complete and without error. This assumption is necessary for training and evaluation and should be somewhat reasonable for central metabolism; however, this assumption is clearly not true, since KEGG is growing.  Thus, there are implications with a using a “gold standard” that is evolving.”

Issue 4:

  1. The Ambiguous dataset is very small (n=353), do you think this drives the performance reported on Table 5. What is the biological meaning of using ambiguous and non-ambiguous sets separately?

Response:

There is no biological meaning for using ambiguous and non-ambiguous sets separately.  Prior KEGG-derived datasets for machine learning had ignored the ambiguous entries.  By including the ambiguous entries, our benchmark dataset better represents KEGG as a whole.  However, their inclusion mandated that we evaluate their effect on the machine learning performance.

Issue 5:

  1. What are the feature names for Table 7?

Response:

These are string representations of the atom coloring features. We have add this to the footnotes:

“The feature name is a string representation of the atom color, i.e., the local chemical neighborhood or subgraph around a specific atom.”

Issue 6:

Some typo needs to be fixed. For example in line 562 'now' should be 'new'.

Response:

Thanks! Fixed.

Reviewer 2 Report

Comments and Suggestions for Authors

In this paper, the authors developed a new KEGG-based benchmark dataset for evaluating metabolomic pathway prediction models. They used the benchmark dataset to train and evaluate different machine learning models, and the best-performing model (XGBoost) provided a 0.8180 F1 and a 0.7933 Matthews correlation coefficient (MCC) for 11 KEGG pathway categories. The addressed question is relevant and interesting to the bioinformatics community. However, the reviewer has the following concerns:

1. The authors excluded the compounds with less than 7 non-hydrogen atoms from the dataset. However, hydrogen metabolism can be central to microbial metabolism and metabolic homeostasis. This approach will also exclude important small molecule metabolites with low atom counts like formic acid, lactic acid, and some amino acids (e.g., glycine), Therefore, this approach risks excluding many biologically relevant compounds, and evaluation using the filtered dataset can be biased. The authors may need to perform a manual inspection to add back key metabolites filtered out in this step. It will also be interesting to evaluate if the proportion of non-hydrogen atoms is associated with the misclassification rate as well.

2. The authors should also present the unweighted averages and standard deviations in Table 5. Weighting by the proportions of pathway categories may introduce some bias because the proportions are different across different datasets, which makes the MCCs not directly comparable. This may also help further address the counter-intuitive full > non-ambiguous > ambiguous test set performance results.

3. The authors claim that the final dataset has improved performance compared with the unfiltered dataset. However, based on Table 6, the standard deviations are quite wide, and the two average MCCs seem to be not significantly different. This is the same for the comparison between different machine learning methods. Figure 7 will benefit from adding pairwise statistical testing results among the violin plots.

4. (Minor) In Figure 1, the text shadow effects can be removed to improve readability. The text descriptions of levels 2-6 and 7-13 of the funnel can be more consistent (e.g., 2-6 can be changed to action items).

Comments on the Quality of English Language

There are some typos throughout the text. For example, line 562 says “a now KEGG-based dataset” and it should be “a new KEGG-based dataset”.

Author Response

Reviewer 2:

In this paper, the authors developed a new KEGG-based benchmark dataset for evaluating metabolomic pathway prediction models. They used the benchmark dataset to train and evaluate different machine learning models, and the best-performing model (XGBoost) provided a 0.8180 F1 and a 0.7933 Matthews correlation coefficient (MCC) for 11 KEGG pathway categories. The addressed question is relevant and interesting to the bioinformatics community.

Response:

We thank the reviewer for their positive comments.

Issue 1:

However, the reviewer has the following concerns:

  1. The authors excluded the compounds with less than 7 non-hydrogen atoms from the dataset. However, hydrogen metabolism can be central to microbial metabolism and metabolic homeostasis. This approach will also exclude important small molecule metabolites with low atom counts like formic acid, lactic acid, and some amino acids (e.g., glycine), Therefore, this approach risks excluding many biologically relevant compounds, and evaluation using the filtered dataset can be biased. The authors may need to perform a manual inspection to add back key metabolites filtered out in this step. It will also be interesting to evaluate if the proportion of non-hydrogen atoms is associated with the misclassification rate as well.

Response:

Pragmatically, any machine learning method for pathway prediction will not be used by itself.  Molecules whose pathway involvement are known, i.e. are identified in KEGG, do not need to be classified by a machine learning method and should not be.  Most of the very small biomolecules are well documented and annotated in KEGG.  So, we are primarily developing a dataset that can train machine learning methods that can be applied to molecules where pathway involvement is not directly known. We have added the following statements to makes these points clearer:

“This is pragmatic, since the prediction of pathway involvement should primarily focus on molecules where direct pathway involvement is not directly known and the pathway involvement of most molecules with less than 7 non-hydrogen atoms is better known.”

Issue 2:

  1. The authors should also present the unweighted averages and standard deviations in Table 5. Weighting by the proportions of pathway categories may introduce some bias because the proportions are different across different datasets, which makes the MCCs not directly comparable. This may also help further address the counter-intuitive full > non-ambiguous > ambiguous test set performance results.

Response:

Respectfully, we disagree with this request.  Weighted averages must be used, since the pathway proportions differ by up to an order of magnitude.  Look at Table S4 and compare “Nucleotide metabolism” pathway proportion of 0.0297 to the “Biosynthesis of other secondary metabolites” pathway proportion of 0.2615 .  Doing a simple average would create too much bias to the pathways with dramatically fewer entries.

Issue 3:

  1. The authors claim that the final dataset has improved performance compared with the unfiltered dataset. However, based on Table 6, the standard deviations are quite wide, and the two average MCCs seem to be not significantly different. This is the same for the comparison between different machine learning methods. Figure 7 will benefit from adding pairwise statistical testing results among the violin plots.

Response:

We agree that the improvement is only 3%.  We could perform a statistical test, but this is subjective to the level of cross-validation performed.  We may have statistical significance, i.e. difference due to random chance being below an alpha of 0.01, with a 1000 cross-validation results.  But we most definitely would have statistical significance if we generated 100,000 cross-validations results.  So the point is really mute.  Thus, we are trying to avoid the p-value vs effect size interpretation of the results. However, the reviewer is correct that we should qualify our interpretation and call out that the difference in performance is small.  We have done this in the following statement:

“Still, the improvement is only (0.7677-0.7454)/0.7454 * 100 ≈ 3%.”

Issue 4:

  1. (Minor) In Figure 1, the text shadow effects can be removed to improve readability. The text descriptions of levels 2-6 and 7-13 of the funnel can be more consistent (e.g., 2-6 can be changed to action items).

Response:

Thanks! Done.

Issue 5:

There are some typos throughout the text. For example, line 562 says “a now KEGG-based dataset” and it should be “a new KEGG-based dataset”.

Response:

Thanks! Fixed.

Reviewer 3 Report

Comments and Suggestions for Authors

The authors proposed an algorithm for creating a dataset of metabolites from KEGG that can be used as a benchmark dataset in the problem of pathway involvement prediction for these metabolites. They also elaborated classification models based on several machine-learning methods and successfully tested these models on this dataset. The methods were described in all details, and results were clearly presented and discussed. All software and algorithms for treating the dataset were made publicly available. I believe the obtained results can be used right away by the scientific community for further applications and development. Overall, I think the manuscript meets all quality requirements, and I don’t have further questions or comments about the study.

Author Response

Reviewer 3:

The authors proposed an algorithm for creating a dataset of metabolites from KEGG that can be used as a benchmark dataset in the problem of pathway involvement prediction for these metabolites. They also elaborated classification models based on several machine-learning methods and successfully tested these models on this dataset. The methods were described in all details, and results were clearly presented and discussed. All software and algorithms for treating the dataset were made publicly available. I believe the obtained results can be used right away by the scientific community for further applications and development. Overall, I think the manuscript meets all quality requirements, and I don’t have further questions or comments about the study.

Response:

We thank the reviewer for their very positive comments!  But we would not have minded any issue, however minor, that the reviewer might have raised.

Round 2

Reviewer 1 Report

Comments and Suggestions for Authors

The authors have address all my comments.